# StEik: Stabilizing the Optimization of Neural Signed Distance Functions and Finer Shape Representation

**Huizong Yang**[1*]   **Yuxin Sun**[1*]   **Ganesh Sundaramoorthi**[2]   **Anthony Yezzi**[1]

[1]Georgia Institute of Technology   [2]Raytheon Technologies   [*]Equal contribution

{huizong.yang,syuxin3,ayezzi}@gatech.edu, ganesh.sundaramoorthi@rtx.com

## Abstract

We present new insights and a novel paradigm (StEik) for learning implicit neural representations (INR) of shapes. In particular, we shed light on the popular eikonal loss used for imposing a signed distance function constraint in INR. We show analytically that as the representation power of the network increases, the optimization approaches a partial differential equation (PDE) in the continuum limit that is unstable. We show that this instability can manifest in existing network optimization, leading to irregularities in the reconstructed surface and/or convergence to sub-optimal local minima, and thus fails to capture fine geometric and topological structure. We show analytically how other terms added to the loss, currently used in the literature for other purposes, can actually eliminate these instabilities. However, such terms can over-regularize the surface, preventing the representation of fine shape detail. Based on a similar PDE theory for the continuum limit, we introduce a new regularization term that still counteracts the eikonal instability but without over-regularizing. Furthermore, since stability is now guaranteed in the continuum limit, this stabilization also allows for considering new network structures that are able to represent finer shape detail. We introduce such a structure based on quadratic layers. Experiments on multiple benchmark data sets show that our new regularization and network are able to capture more precise shape details and more accurate topology than existing state-of-the-art.[1]

## 1   Introduction

Implicit neural representations (INR) [1]–[17], which are neural network representations for implicit representations of signals (e.g., shape, images), have recently become a powerful tool for modeling shape in learning based frameworks for surface reconstruction tasks [1]–[3], [10]–[17] in computer vision and graphics. INRs typically represent a shape as the zero level set of its corresponding signed distance function (SDF), which is represented with a neural network (e.g., a multi-layer perceptron - MLP). To learn an INR, one minimizes a loss consisting of a data fidelity term (e.g., fidelity to known points on the surface, i.e., a point cloud, for the point cloud to surface reconstruction task [1]–[3], [10]–[17]) and regularization terms. A regularization term used is the eikonal loss [11], which constrains the neural representation to be an SDF. While existing methods have shown the ability to recover complex scenes and objects, in many cases as datasets become more complex, finer scale geometric and topological structures may not always be recovered.

In this paper, we show that the continuum limit of the optimization of neural SDFs as the network representation power increases (to recover finer scale shape features) can be unstable due to the eikonal loss. This limits recovery of fine shape details and/or can lead to convergence to sub-optimal

---

[1]Code: `https://github.com/sunyx523/StEik`

37th Conference on Neural Information Processing Systems (NeurIPS 2023).

local minima resulting in gross errors. We provide a theoretical framework based on geometric PDEs and PDE stability analysis [18]–[21], which has recently been proven to be a powerful tool for analyzing deep learning optimization [22]–[24], to study the optimization of the eikonal loss. Within this framework, we show how other terms (such as the normal constraint [10] and divergence loss [14]) in state-of-the-art proposed for various end goals can also have a stabilizing effect, giving new justification for such terms. However, we show (both in theory and empirically) that such terms, which we show are penalties on surface area and mean curvature, can over-regularize the resulting surface, thus failing to capture fine details of the shape (e.g., thin structures). Based on geometric PDEs, we show how to construct a regularization term that stabilizes the PDE, but does not over-smooth the surface.

Furthermore, stabilizing the eikonal loss optimization with our new regularization enables the use of new neural networks with higher representation power that can capture finer scale details of shape, without suffering from the destabilizing effects of the continuum PDE. We demonstrate this point by proposing a new network structure for neural SDFs, based on quadratic layers [25]–[31], which have thus far been applied mainly to classification tasks. Unlike compositions of linear layers that preserve linearity, compositions of quadratic functions are not quadratic, but can be a higher-order polynomial. Thus, we can represent shape with finer piece-wise Taylor approximations, resulting in finer shape representation with fewer network parameters than state-of-the-art [14].

Our contributions are: **1.** We provide a theoretical framework, based on geometric PDEs, to study the optimization of neural SDFs. In particular, we use PDEs to analyze the eikonal loss, and show it can be unstable. This theory sheds light on the design of neural SDFs and serves as a framework to design new methods. **2.** We use this framework to analyze existing terms proposed in the literature, and show how some can provide a stabilizing effect, even though they were motivated for other end goals. This provides new theoretical justification for these methods while also providing a rigorous analysis of their limitations. **3.** We use geometric PDEs to propose a new loss regularization, i.e., second order derivative in the normal direction, that avoids over-regularization while stabilizing the eikonal loss. **4.** We propose the use of quadratic layers for neural SDFs, which provide an arbitrary order piece-wise polynomial approximation of the shape. This provides a finer shape representation than existing art, without being subject to the instabilities of the eikonal loss. **5.** We provide an extensive benchmark comparison to state of the art on three datasets: the Surface Reconstruction Benchmark [32], ShapeNet [33], and large scene reconstruction [12]. We demonstrate that our method consistently improves state-of-the-art, especially on challenging geometries and topologies.

We call our method *StEik* for stabilizing the eikonal equation. Note that while we benchmark our methods on the task of surface recovery from point cloud data, our theory/methods can be applied to other problems that aim to recover neural SDFs.

## 2   Related Work

**Shape implicit neural representations (INRs):** Traditional approaches for representing 3D shapes such as meshes pose difficulties in integrating them with deep learning methods. Deep learning approaches operating directly on traditional representations have limited quality and flexibility. Recently, neural networks have been proposed to represent shapes and scenes using signed distance functions (SDF) [1], [10]–[16] or an occupancy functions [2], [3], [17]. They have proven more convenient and accurate than traditional representations within deep learning-based solutions. DeepSDF [13] was the first to introduce the use of SDFs in INRs, and was used to represent a collection of shapes. It regresses on ground truth SDFs. In many applications, such SDF ground truth is difficult to obtain. Thus, some methods learn INRs directly from raw data, e.g., point clouds (from range scanners) or 2D images (e.g., [3], [16]) in multi-view reconstruction applications.

SAL [1] aims to recover SDFs from point clouds. SALD [10] further improves SAL by incorporating surface normal data. IGR [11] proposes a loss function based on the eikonal equation, which helps regularize the learned function towards an SDF. FFN [17] and SIREN [12] introduce high frequencies into their architecture to avoid bias towards low-frequency solutions in different ways. FFN uses a ReLU MLP that is paired with a Fourier feature layer. SIREN uses the sine activation. Recently, DIGS[14] improves the performance of SIREN on shape representation tasks by proposing a soft constraint on the divergence of the gradient field and proposing a new initialization method. DiGS is motivated by avoiding the use of normal data, which is not available for many applications.

Approaches above recovering SDFs use the eikonal constraint in training, which we show limits to an unstable PDE, causing artifacts that limit recovery of fine details or convergence to sub-optimal local minima. We provide a theoretical framework to understand this instability and explain how some existing approaches can unknowingly mitigate this instability, however, producing an undesirable over-regularizing effect. Our framework enables us to design a new regularizer that stabilizes the eikonal term while recovering finer geometric details (without over-regularizing).

**Quadratic Deep Neural Networks:** Our new regularization enables us to use new neural networks for SDFs to represent finer shape details without suffering from destabilizing effects of the eikonal term that become more prominent in higher capacity networks. We use quadratic layers in INRs, which is novel, to illustrate this point. Quadratic Deep Neural Networks (QDNNs), proposed back in 1990s [25], [26], have been recently used to enhance the learning capability of Deep Neural Networks (DNNs) [27]–[31]. Rather than linear functions used in conventional linear layers, a quadratic function is used. Since compositions of quadratic functions can be higher-order polynomials, such QDNNs can represent piece-wise polynomial functions. Thus, QDNNs have better model efficiency because they can approximate polynomial decision boundaries using smaller network depth/width. However, the improvement is limited when it is applied to Convolution Neural Networks (CNN) [34]–[39]. In contrast, we demonstrate that using quadratic neurons in MLPs for representing shapes as implicit functions is highly effective.

# 3 Theory and Analysis of the Stability of Neural SDF Optimization

In this section, we present geometric PDEs as a framework for analyzing the continuum limit of neural SDF optimization, show the instability in the eikonal loss, and show how existing neural SDF approaches can mitigate the instability. This serves as a framework for our new methods in Section 4. To analyze stability of the optimization/PDE, we use spectral methods from numerical PDEs and Von Neumann analysis. Our exposition is self-contained, however, for a more detailed introduction, see Ch. 4 of [40] and Ch. 4.3 of [41], and for a tutorial on stability analysis in the context of deep networks, see [24].

## 3.1 PDE as the Continuum Limit of Neural SDF Optimization

Let $u : \Omega \subset \mathbb{R}^n \to \mathbb{R}$ be the function that is evolving in the continuum (e.g., level set representation; the hyper-surface of interest is the zero level set, i.e., $\{x \in \Omega \ : \ u(x) = 0\}$). This is the continuum limit of the typical neural SDF evolutions. Suppose the loss of interest (defined on $u$) is $L$. Then the gradient descent is given by the PDE:

$$\frac{\partial u}{\partial t} = -\nabla L(u), \tag{1}$$

where $t$ is the artificial parameter of the evolution (the continuum equivalent of the iteration index), $\nabla L$ satisfies the relation $\delta L \cdot \delta u = \langle \delta u, \nabla L(u) \rangle_{\mathbb{L}^2}$, and the latter expression is the $\mathbb{L}^2$ inner product between the gradient and the (infinite dimensional) perturbation of $u$, $\delta u$. Note that while we analyze stability of gradient descent, our analysis also applies to second order optimizers (e.g., Nesterov momentum) as such optimizers do not change stability properties [18], [24]. Suppose now that $u$ is parameterized by $\theta$, denoted $u_\theta$, as in neural SDFs. We compute the projected gradient descent of the loss with respect to the parameters $\theta$. Note that if we wish to perturb $u$ according to a perturbation $\delta\theta$, then $\delta u = \frac{\partial u}{\partial \theta} \cdot \delta\theta$. Therefore,

$$\delta L \cdot \delta\theta = \int_\Omega \nabla L(u)(x) \frac{\partial u}{\partial \theta}(x) \cdot \delta\theta \, \mathrm{d}x = \int_\Omega \nabla L(u)(x) \frac{\partial u}{\partial \theta}(x) \, \mathrm{d}x \cdot \delta\theta.$$

Thus, the projected gradient descent in parameter-space is

$$\frac{\mathrm{d}\theta}{\mathrm{d}t} = -\int_\Omega \nabla L(u)(x) \frac{\partial u}{\partial \theta}(x) \, \mathrm{d}x. \tag{2}$$

The corresponding PDE evolution of the neural representation (in function space) is

$$\frac{\partial u}{\partial t} = \frac{\partial u}{\partial \theta} \frac{\mathrm{d}\theta}{\mathrm{d}t} = -\sum_i \frac{\partial u}{\partial \theta_i} \left\langle \nabla L(u), \frac{\partial u}{\partial \theta_i} \right\rangle_{\mathbb{L}^2}, \tag{3}$$

where $\mathcal{B} = \left\{ \frac{\partial u}{\partial \theta_i} \right\}_i$ is a basis for the sub-space of the tangent space of function representations that is spanned by the parameterization of the network (e.g., neural SDF). The evolution above is simply a projection of the continuum gradient of the loss onto the basis of the tangent space formed by the neural representation.

Note that as the neural network representation gains more representational power (more capacity to represent finer scale and more divers shapes), the basis $\mathcal{B}$ approaches spanning the entire tangent space of functions, i.e., in $\mathbb{R}^n$, and hence the projected PDE approaches the full PDE (1). Therefore, analyzing the unconstrained PDE (1) gives insight into the neural representation. In the next sub-sections, we will focus on the notion of *stability* of the PDEs, which impacts the accuracy of the neural representation.

## 3.2 PDE Stability Analysis: Theoretical Framework for Analysis of Neural SDF Optimization

Current approaches for learning a neural signed distance function minimize a loss that consists of a data fidelity term and regularization. Regularization aims to keep the representation close to a signed distance function, and can also include terms that regularize the underlying shape (e.g., to keep the shape smooth). In this sub-section, we will focus on the eikonal loss that is part of the regularization. A necessary condition for a signed distance function is that it satisfies the eikonal PDE and thus the eikonal loss penalizes deviation from that constraint:

$$|\nabla u(x)| = 1, \quad x \in \Omega, \quad \implies L_{\text{eik}}(u) = \frac{1}{2} \int_\Omega ||\nabla u(x)| - 1|^p \, \mathrm{d}x, \quad (4)$$

where $p = 1$ or $p = 2$ for a $L^1$ or $L^2$ loss, respectively and $\nabla u$ is the spatial gradient.

We claim that the gradient descent PDE for the eikonal loss maybe *unstable* at some space-time locations. By stability, we mean that the solution of the PDE converges as $t \to \infty$. By Von Neumann analysis [40], if the homogeneous component of the linearization is non-zero, and the evolution in the frequency (Fourier) domain has an unbounded amplifier, the PDE is unstable. We use Von Neumann analysis to show that the gradient descent PDE of the eikonal loss is unstable. By arguments in the previous section, this means that as the representation of the power of the neural SDF increases, the optimization can become unstable. The gradient descent PDE for the Eikonal loss is

$$\frac{\partial u}{\partial t} = \nabla \cdot (\kappa_e \nabla u), \quad \kappa_e(x) = \begin{cases} \mathrm{sgn}[1 - |\nabla u(x)|]/|\nabla u(x)| & p = 1 \\ |\nabla u|^{-1} - 1 & p = 2 \end{cases}, \quad (5)$$

where sgn is the sign function. The local linearization of this equation is obtained by treating $\kappa_e$ as constant, which is true locally; this results in the linearization:

$$\frac{\partial u}{\partial t} = \kappa_e \Delta u, \quad (6)$$

where $\Delta$ denotes Laplacian, and note $\kappa_e$ can be positive or negative. When $\kappa_e < 0$, the process is a backward diffusion, which is ill-posed and therefore fundamentally unstable, regardless of the numerical implementation scheme to be used. To see this, we may compute the spatial Fourier transform of the above equation, which yields:

$$\frac{\partial \hat{u}}{\partial t}(t, \omega) = -\kappa_e |\omega|^2 \hat{u}(t, \omega) \implies \hat{u}(t, \omega) \propto e^{-\kappa_e |\omega|^2 t}, \quad (7)$$

where $\omega = (\omega_1, \ldots, \omega_n)$ is the frequency variable, and $\hat{u}$ is the Fourier transform of $u$. Notice that when $\kappa_e < 0$, the process diverges and so is unstable. Therefore, the projected gradient descent PDE of the Eikonal loss when $u$ is represented with a (parametric) neural representation can become unstable as the representational power of the neural SDF increases (approaching the continuum limit).

One may wonder, if the optimization of the Eikonal loss is unstable, why the network optimization seems to converge. There may be several reasons for this. Firstly, since $\kappa_e$ can be positive or negative at certain locations, the PDE could go from unstable to stable and even oscillate between these two states without fully blowing up. However, this can cause irregularities in the evolution and recovered shape (see Figure 1). Second, due to the finite parameterization of neural representations, networks with less capacity may project to a flow that annihilates some of the unstable components. Lastly, as we will see in the next sub-section through analysis, various regularization terms introduced (for other purposes) can have a stabilizing effect. Nevertheless, these approaches can limit the representational power of the network to represent fine-scale shape details. Our approaches in the next section, built upon our theory, stabilize while allowing more complex networks to have finer shape representation.

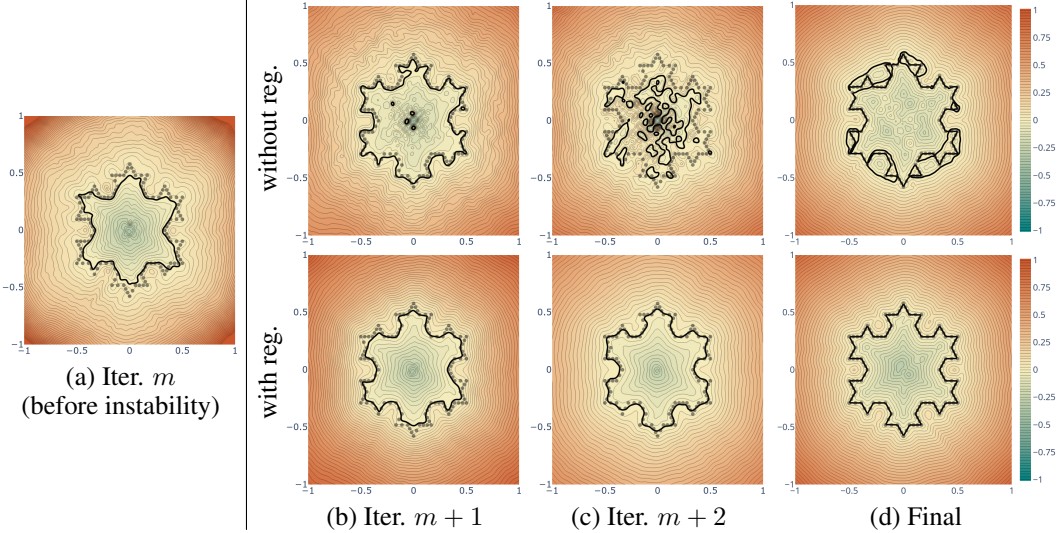

Figure 1: Visual demonstration of the eikonal instability in the INR. (a) shows the level set at the iteration, $m$, before an instability of the non-regularized SDF optimization. [*Top row*]: shows the level set at various subsequent iterations (b)-(d) of the continued non-regularized SDF evolution. [*Bottom row*]: shows the level set at various iterations of the SDF evolution when adding our proposed regularization (directional divergence) after (a).

## 3.3 PDE Stability Analysis of Existing Neural SDF Representations

We now use our theory to analyze existing methods. In [14], a regularization term is added to the loss function for training neural SDFs; the loss (called the divergence loss) is as follows:

$$L_{\text{div}}(u) = \int_{\Omega \setminus \Omega_0} |\Delta u(x)|^p \, dx, \tag{8}$$

where $\Omega_0$ are points on the ground truth surface (e.g., points of a point cloud or the zero level set of the ground truth). The authors observe empirically that the Laplacian of a SDF is close to zero and thus this is added as a constraint. Although we show in the next section that this is not always or only partially true, we will now show that this term has another beneficial property, i.e., that it stabilizes the instability of the eikonal loss gradient descent. The gradient descent PDE for the sum of the above divergence loss and the eikonal loss ($\alpha_e L_{\text{eik}} + \alpha_d L_{\text{div}}$, where $\alpha_e, \alpha_d > 0$ are weights) is

$$\frac{\partial u}{\partial t} = \alpha_e \nabla \cdot [\kappa_e \nabla u] - \alpha_d \begin{cases} \Delta[\text{sgn}(\Delta u)] & p = 1 \\ \Delta[\Delta u] & p = 2 \end{cases} \tag{9}$$

which is a fourth-order PDE. Note that in implementations, one would have to approximate the sign function with a differentiable approximation. We will assume $\text{sgn}(x) = 2\sigma(x) - 1$, where $\sigma$ is the sigmoid function, i.e., the key property is that the approximation is positively sloped near the origin, and close to a constant away from the origin on either side. Note that the stability of the PDE is typically dominated by the highest-order terms, which in the above case is stable. To see this, we linearize the first term as done previously (assuming $\kappa_e$ is constant, and approximating sign as linear near the origin and constant elsewhere). In this case,

$$\Delta[\text{sgn}(\Delta u)](x) \approx \begin{cases} \kappa_d \Delta[\Delta u](x) & \Delta u(x) \approx 0 \\ 0 & |\Delta u(x)| \gg 0 \end{cases},$$

where $\kappa_d > 0$ is the slope of the sign approximation at zero. Therefore, in both $p = 1$ and $p = 2$, the linearization of the PDE (near $\Delta u = 0$ for $p = 1$ and everywhere for $p = 2$) is given by

$$\frac{\partial u}{\partial t} = \alpha_e \kappa_e \Delta u - \alpha_d \kappa_d \Delta[\Delta u] = \alpha_e \kappa_e \sum_{j=1}^{n} \frac{\partial^2 u}{\partial x_i^2} - \alpha_d \kappa_d \sum_{j,k=1}^{n} \frac{\partial^4 u}{\partial x_j^2 \partial x_k^2}. \tag{10}$$

Computing the spatial Fourier transform of the above linearized equation yields:

$$\frac{\partial \hat{u}}{\partial t}(t, \omega) = - \left[ \alpha_e \kappa_e |\omega|^2 + \alpha_d \kappa_d |\omega|^4 \right] \hat{u}(t, \omega) = A(w) \hat{u}(t, \omega) \implies \hat{u}(t, \omega) \propto e^{A(\omega)t}. \quad (11)$$

Note that in any local approximation of $\kappa_d$ with a constant, $\kappa_d > 0$. Thus, regardless of the sign of $\kappa_e$, so long as $\alpha_d$ is chosen large enough, the set in which $A(\omega)$ is positive can be minimized, and so the process is stable. Thus, besides aiming to enforce the empirically observed property that the Laplacian of the neural SDF is close to zero, that term also adds stability to the neural SDF optimization, adding a regularizing effect.

In several works, a term is included to penalize the deviation between the normal to the SDF and the ground truth normal direction to the surface (or point cloud), which provides further constraints on the recovered SDF. In some problems this ground truth data is available. In addition to serving as an additional constraint, for particular forms of that constraint [11], we show that term can stabilize the eikonal term. The normal constraint is given by the loss:

$$L_{norm.}(u) = \int_{\Omega_o} |\nabla u(x) - N_{gt}|^p \, dx, \quad (12)$$

where $\Omega_o$ are points on the ground truth surface. The gradient descent of this term is given by

$$\frac{\partial u}{\partial t} = \nabla \cdot [\kappa_n (\nabla u - N_{gt})], \quad \kappa_n = \begin{cases} |\nabla u - N_{gt}|^{-1} & p = 1 \\ 1 & p = 2 \end{cases}, \quad (13)$$

which includes a forward diffusion, which if the weight on this term is chosen larger than $-\alpha_e \kappa_e$, would stabilize the (unstable) backward diffusion of the eikonal loss.

# 4 New Shape Regularization and Representation for Finer Neural SDFs

## 4.1 A New Stabilizing Term Without Over-Regularization

We introduce a new stabilizing term for eikonal loss. To motivate our approach, we first shed some more insight into the divergence loss (penalty on the Laplacian of $u$, the SDF representation). We first recall a fact from differential geometry. For a hyper-surface in $\mathbb{R}^n$, the *mean curvature $H$* of the hyper-surface measures the average turning of the unit normal with respect to $n$ principal directions of the surface [42]. We avoid precisely defining the mean curvature due to the lengthy technical details needed, and refer the reader to [42]. Of particular interest is expressing the mean curvature of a surface in terms of its level set embedding. If $u$ is a level set function, then the mean curvature of the level sets can be written as the divergence of the normal vector field to the level sets [43], i.e.,

$$H = \text{div} \left( \frac{\nabla u}{|\nabla u|} \right). \quad (14)$$

Note that if $u$ is a signed distance function, it satisfies the eikonal equation and thus the mean curvature is the Laplacian of the SDF, i.e., $H = \Delta u$. Hence, for arbitrary shapes, the Laplacian of an SDF is the mean curvature of the level sets, which is not always close to zero. If we would like to represent shapes with fine detail and complex curvatures, penalizing the Laplacian of $u$ in the loss would not necessarily be beneficial, although the term stabilizes the eikonal loss.

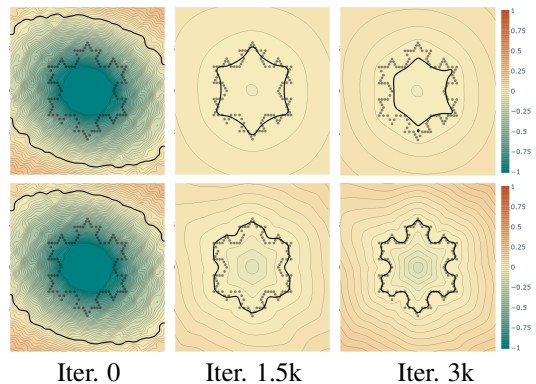

| Iter. 0 | Iter. 1.5k | Iter. 3k |

Figure 2: Illustration of the ability of our new regularization to capture fine-scale details of shape while still stabilizing the optimization. Without a penalty on the mean curvature, our directional divergence term restores the shape more quickly and captures fine details (bottom). On the other hand, the full divergence term (top) unnecessarily penalizes the mean curvature of the level sets, resulting in over-smoothness. Note the dark black lines represent the zero level set (lighter ones indicate other level sets). Ground truth is a snow-flake like shape (dotted gray). Note that both divergence terms avoid instabilities, but our proposed one recovers the correct curvatures of the shape.

However, we note that there is a component of the Laplacian of a SDF that is zero. Indeed, if we compute the gradient of both sides of the eikonal equation ($|\nabla u(x)| = 1$), we obtain that

$$0 = D^2 u(x) \cdot \frac{\nabla u(x)}{|\nabla u(x)|} = D^2 u(x) \cdot \nabla u(x), \tag{15}$$

where $D^2 u(x)$ indicates the Hessian of the SDF, and the dot indicates matrix-vector multiplication. Note that the above quantity dotted with $\nabla u$ is the second derivative of $u$ in the normal direction of the level sets, which is a component of the full Laplacian of $u$. Hence, we introduce a new loss term as a replacement for the penalty on the full Laplacian, which we refer to as *Laplacian normal* regularization or *directional divergence*:

$$L_{\text{L. n.}}(u) = \int_\Omega |\nabla u(x)^T D^2 u(x) \cdot \nabla u(x)| \, dx. \tag{16}$$

This loss enforces the constraint in SDFs that the second derivative in the normal direction is zero, without enforcing unwanted smoothness by penalizing the fine detail (points of high mean curvature) of the level sets. This will lead to a fourth-order (non-linear) PDE for its gradient descent. The gradient descent PDE includes a term that is $-\Delta[\Delta u]$, an isotropic fourth order term, which from the previous analysis would stabilize the lower order eikonal instability. Although the full flow only regularizes in the normal direction, over the evolution it regularizes over other directions as the normal vector changes direction, killing the eikonal instability.

**New Training Loss:** We combine the new stabilizing term with the loss function used in SIREN [12] without the normal constraint (for more applicability) to form our proposed training loss:

$$L = \alpha_e L_{\text{eik}} + \alpha_m L_{\text{manifold}} + \alpha_n L_{\text{non manifold}} + \alpha_l L_{\text{L.n.}},$$

$$L_{\text{manifold}} = \int_{\Omega_0} |u(x)| \, dx, \quad L_{\text{non manifold}} = \int_{\Omega \setminus \Omega_0} \exp\left(-\alpha |u(x)|\right) dx, \tag{17}$$

where $\alpha, \alpha_e, \alpha_n, \alpha_m, \alpha_l > 0$ are hyper-parameters, and $\Omega_0$ are known points on the surface of interest (e.g., point cloud data). $L_{\text{manifold}}$ penalizes surface points away from the zero level set. $L_{\text{non manifold}}$ penalizes points not on the surface of interest from being close to the zero level set. We use $p = 1$ for the Eikonal loss, the same as in SIREN [12] and DiGS [14]. For $\alpha_l$ we use the annealing strategy as in DiGS [14]. See supplementary for details.

## 4.2 A New Representation for Finer Shape Representation in Neural SDFs

We now introduce a new neural network representation for SDFs, motivated by our result that allows stabilizing the eikonal loss even when the representational power of the network increases. Note in a ReLu MLP, the network represents a piecewise-linear function. Activations partition the domain where various linear approximations are used. To capture finer details of shape (without resorting to heavy linear networks), it is natural to leverage more general Taylor series (quadratic and beyond) approximations to capture the curvature of the shape. Motivated by this observation, we propose to use quadratic layers rather than linear layers. Notice that the composition of a quadratic with a quadratic function is a quartic function, and thus composing quadratic layers many times can approximate any desired order of a Taylor series, even without the use of activations. We still use activations, however, to partition the domain into regions where different Taylor approximations are used. Without stabilizing the eikonal term in the optimization, such finer-scale representations become unstable; thus, our regularization plays a crucial role. Note quadratic layers have been proposed for neural networks [30]; however, proposing them for shape representation in neural SDFs is novel to the best of our knowledge. Note also that SIREN [12] uses a sinusoidal activation to obtain a representation beyond piecewise linear in ReLu MLPs; in that representation, however, the activation serves to both partition the domain in pieces, and represent each piece with more complex (polynomial) functions. Quadratic layers allow more complex (polynomial) functions in the pieces, without overloading the activation with both partitioning the domain and more complex function representation.

As in [30], we define a quadratic layer using the following representation:

$$\mathbf{a}(\mathbf{x}) = (W_1 \mathbf{x} + \mathbf{b_1}) \circ (W_2 \mathbf{x} + \mathbf{b_2}) + W_3 \mathbf{x}^2 + \mathbf{b_3}, \tag{18}$$

where $W_j \in \mathbb{R}^{m_1 \times m_2}$, $\mathbf{x} \in \mathbb{R}^{m_2}$ is the input vector, $\mathbf{x}^2$ is the element-wise square, $\mathbf{b_j} \in \mathbb{R}^{m_2}$ are biases, and $\circ$ denotes the element-wise product. We replace the linear neurons in the SIREN [12] network with quadratic neurons to obtain a high-order expression for the signed distance function. For implementation, we use the combination of three linear layer modules in PyTorch.

# 5 Experiments

We now demonstrate the effectiveness of our method on the task of surface reconstruction from point clouds. For all the experiments in this section, we follow the same mesh generation procedure and evaluation setting as the state-of-the-art method DiGS [14]. We experiment on three benchmarks: Surface Reconstruction Benchmark (SRB) [32], ShapeNet [33], and the Scene Reconstruction Benchmark [12]. We use a network with 5 hidden layers and 128 hidden channel for SRB and ShapeNet, and we use 8 hidden layers, and 256 channels for scene reconstruction. The number of training iterations is the same as in DiGS [14], 10k for SRB and ShapeNet, and 100k for scene reconstruction. We provide all of the training details in the supplementary.

## 5.1 Surface Reconstruction Benchmark (SRB)

SRB consists of 5 noisy range scans and each contains point cloud and normal data. We compare our method against the current state-of-the-art methods on this benchmark without using normal data (as in [14] as normal data may be difficult to obtain). Results are shown in Table 1. We report the Chamfer ($d_C$) and Hausdorff ($d_H$) distances between the reconstructed meshes and the ground truth meshes. Furthermore, we report their corresponding one-sided distances ($d_{\vec{C}}$ and $d_{\vec{H}}$) between the reconstructed meshes and the input noisy point cloud, which measures how much the reconstruction overfits noise in the input. Results show that StEik is better than SoTA methods on

| Method | GT | | Scans | |
| | $d_C$ | $d_H$ | $d_{\vec{C}}$ | $d_{\vec{H}}$ |
|---|---|---|---|---|
| IGR wo n | 1.38 | 16.33 | 0.25 | 2.96 |
| SIREN wo n | 0.42 | 7.67 | 0.08 | **1.42** |
| SAL[1] | 0.36 | 7.47 | 0.13 | 3.50 |
| IGR+FF[17] | 0.96 | 11.06 | 0.32 | 4.75 |
| PHASE+FF[17] | 0.22 | 4.96 | **0.07** | 1.56 |
| DiGS[14] | 0.19 | 3.52 | 0.08 | 1.47 |
| Our StEik | **0.180** | **2.800** | 0.096 | 1.454 |

Table 1: Quantitative results on the Surface Reconstruction Benchmark[32] using only point data (no normals).

the ground truth metrics, but can slightly overfit the noisy input due to the fine representation property of our method. The improvement is not so dramatic compared to DiGS [14] because this SRB is a relatively easy task without many thin structures and complex structures, and DiGS [14] already has a good performance. However, we still achieve a better result with 25% fewer parameters than DiGS.

## 5.2 ShapeNet

We evaluated our method on a preprocessed subset [44], [45] of ShapeNet [33], which consists of 20 shapes in each of 13 categories with only surface point data. Note points are sampled from the shapes (as in [44]) to simulate point clouds. We compare StEik against the current state-of-the-art methods on this dataset without using normal data and report the results in Table 2. As criteria for the benchmark, we consider the Intersection over Union (IoU) and Chamfer Distance between the reconstructed shapes and the ground truth shapes. The Intersection over Union (IoU) captures the accuracy of the predicted occupancy function, while the Chamfer Distance captures the accuracy of the predicted surface. Under both metrics, StEik outperforms all other methods by a large margin. This demonstrates that StEik is particularly effective for reconstructing thin structures. A visual example is shown in Figure 3 (see supplementary for more).

| Method | squared Chamfer ↓ | | | IoU ↑ | | |
| | mean | median | std | mean | median | std |
|---|---|---|---|---|---|---|
| SIREN wo n | 3.08e-4 | 2.58e-4 | 3.26e-4 | 0.3085 | 0.2952 | 0.2014 |
| SAL[1] | 1.14e-3 | 2.11e-4 | 3.63e-3 | 0.4030 | 0.3944 | 0.2722 |
| DiGS[14] | 1.32e-4 | 2.55e-5 | 4.73e-4 | 0.9390 | 0.9764 | 0.1262 |
| **Ablation (of Regularizations & Linear vs Quad Layers)** | | | | | | |
| Lin+$L_{\text{L. n.}}$ | 1.71e-4 | 1.23e-5 | 1.20e-3 | 0.9586 | 0.9809 | 0.0993 |
| Qua+$L_{\text{div}}$ | **5.45e-5** | 1.05e-5 | 3.60e-4 | 0.9593 | **0.9852** | 0.1130 |
| Our StEik (Qua+$L_{\text{L. n.}}$) | 6.86e-5 | **6.33e-6** | **3.34e-4** | **0.9671** | 0.9841 | **0.0878** |

Table 2: Quantitative results on the ShapeNet[33] using only point data (no normals).

**Ablation Study:** Below the middle line in Table 2, we study the effectiveness of each of our novel contributions (the Laplacian normal regularization and quadratic layers). On 5 out of 6 metrics,

our normal Laplacian regularization using linear networks outperforms DiGs (which uses linear networks), showing the utility of the new regularization. On 4 out of 6 metrics, our normal Laplacian regularization out-performs the standard Laplacian regularization using quadratic networks, again showing the utility of our new regularization separately. Note that in both cases the metrics where the normal Laplacian normal performs worse are just slightly worse compared to the amount of increase in the other metrics. On all 6 metrics, the quadratic network using the same Laplacian regularization as DiGs out-performs DiGs, showing the utility of quadratic networks alone. Note that each of our contributions, i.e., Laplacian normal regularization and quadratic layers, separately show increase performance against DiGs (except one metric in the linear case) even though the hyper-parameters were not optimized in the approaches compared against DiGS.

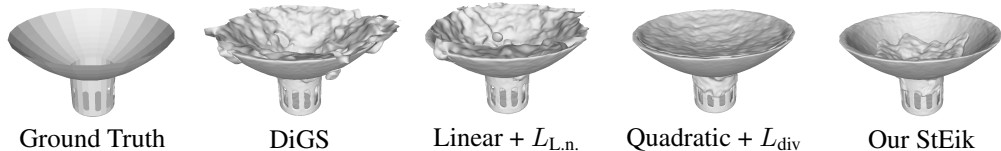

| Ground Truth | DiGS | Linear + $L_{\text{L.n.}}$ | Quadratic + $L_{\text{div}}$ | Our StEik |

Figure 3: Example Visual results on ShapeNet [33]: We manifest the effectiveness of the new regularization and the new representation of Neural SDFs independently. Furthermore, the combination of two modules demonstrates an extra improvement. See supplement for more visual results.

## 5.3 Scene Reconstruction

In Figure 4, we show the reconstruction of a room scene point cloud from roughly 10M points and compare our method with DiGS[14], the current SoTA method without normals. This is the same scene used in [12] and contains many thin features that are difficult to reconstruct. The surface produced by DiGS is over-smoothed so that the thin structures like picture frames and sofa legs are not recovered, while in StEik those fine details are recovered.

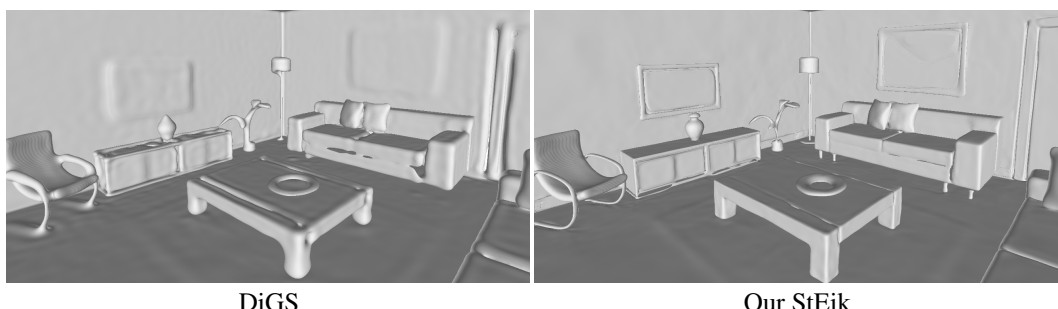

| DiGS | Our StEik |

Figure 4: Visual results on the Scene Reconstruction Benchmark using only point data (no normals).

## 5.4 Timing Performance and Model Size

Table 3 compares the training time of one iteration and the number of parameters of DiGS [14] and our method. The evaluation is performed on a single Nvidia Tesla A100 GPU. The setting above the line is for SRB [32] and ShapeNet [33] experiments. The setting below the line is for the scene reconstruction experiment [12]. We achieve better performance than DiGS [14] with only 3/4 the number of parameters. There is an increase in training time (per iteration) for StEik compared to DiGS due to the extra computation cost introduced by quadratic neurons. Note that our $L_{L.n}$ regularization is computationally less expensive

| Method | Structure | Runtime | Parameters |
|---|---|---|---|
| DiGS | 5×256 | 37.86ms | 0.26M |
| Lin+$L_{\text{L.n.}}$ | 5×256 | 32.52ms | 0.26M |
| Qua+$L_{\text{div}}$ | 5×128 | 50.92ms | 0.20M |
| Our StEik | 5×128 | 42.20ms | 0.20M |
| DiGS | 8×512 | 63.28ms | 1.84M |
| Lin+$L_{\text{L.n.}}$ | 8×512 | 50.90ms | 1.84M |
| Qua+$L_{\text{div}}$ | 8×256 | 100.27ms | 1.39M |
| Our StEik | 8×256 | 80.62ms | 1.39M |

Table 3: Network structure, speed, and size comparison. 5×256 means 5 layers with 256 neurons.

than $L_{div}$ (proposed in DiGS) as $L_{L.n}$ can be expressed as $\frac{1}{2}\nabla_x\left[\langle\nabla u(x),\nabla u(x)\rangle\right]$, which can be computed with a single back-propagation call. In contrast there is no such simpler expression for the Laplacian.

## 5.5 Limitations

Due to the lack of efficient implementation of quadratic layers in the deep learning libraries, the increase in training time is not negligible. Indeed, there is no native Pytorch implementation of a quadratic layer, and so we currently stack linear layers. As quadratic layers become more frequently used, we expect that a native Pytorch implementation become available, which would address the current drawback. In addition, there is still improvement space for reconstruction results, as in some cases the surface is not perfectly recovered.

## 6 Conclusion

We showed that stability is an important consideration in the design of neural SDF representations. We showed that the eikonal loss can result in instabilities that can cause artifacts in both the optimization and the recovered shape, or even converge to sub-optimal local minima. Our theory allows for understanding the instability and existing methods for neural SDFs in a common framework. Our framework enabled the construction of a new regularization term for neural SDFs that stabilizes the instability while avoiding over-regularization. The regularization enabled us to consider finer shape representations with neural SDFs that are piecewise polynomial while stabilizing the eikonal term. Empirical results validated our theoretical findings. This work opens up the possibility of exploring a broader range of geometric regularizations that naturally arise from PDEs, and the possibility of exploring new finer-scale network representations.

## Acknowledgements

Supported in part by Army Research Office (ARO) W911NF-22-1-0267 and by the Intelligence Advanced Research Projects Activity (IARPA) via Department of Interior/ Interior Business Center (DOI/IBC) contract number 140D0423C0075. The U.S. Government is authorized to reproduce and distribute reprints for Governmental purposes notwithstanding any copyright annotation thereon. Disclaimer: The views and conclusions contained herein are those of the authors and should not be interpreted as necessarily representing the official policies or endorsements, either expressed or implied, of IARPA, DOI/IBC, or the U.S. Government.

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
