# OpenReview forum: "StEik: Stabilizing the Optimization of Neural Signed Distance Functions and Finer Shape Representation"
_NeurIPS.cc/2023/Conference — NeurIPS 2023 poster_

### Official Review · Reviewer_urXf · 2023-07-01

**Soundness:** 3 good
**Presentation:** 3 good
**Contribution:** 3 good
**Rating:** 5
**Confidence:** 3

**Summary:**

The authors demonstrate that the eikonal equation which is often used for representing 3D shapes leads to training instabilities. Based on this finding the authors propose a new regularizer for representing 3D shapes based on the eikonal equation. This regularizer is based on the hessian of the SDF function. Moreover, the authors propose to use quadratic layers instead of linear layers.

**Strengths:**

The eikonal equation indeed leads to training instability. The solution based on a second order regularizer seems sensible. The empirical results seem to support the regularizer and architecture choice.

**Weaknesses:**

- In table 3, why does the runtime decrease when adding the new regularizer which is based on the hessian? I would assume that this impacts the runtime negatively. Can the authors elaborate on this?
- How does the number of parameters of the methods in tab 1 compare to each other?


**Questions:**

see weaknesses

---

> ### Author Rebuttal · Authors · 2023-08-10
>
> **Q1** *In table 3, why does the runtime decrease when adding the new regularizer which is based on the hessian? I would assume that this impacts the runtime negatively. Can the authors elaborate on this?*
>
> **->** The major savings comes from the number of back-propagation calls.  In our approach, we only need to compute derivatives of scalar functions in comparison to the divergence term. Since $D^2(x)\nabla u(x)$ can be computed conveniently as half the derivative of $\langle \nabla  u(x), \nabla  u(x) \rangle$ with respect to $x$, we can compute $L_{L.n} = \langle D^2(x)\nabla u(x), \nabla u(x)\rangle$ by a single back-propagation call. In contrast, there is no simpler expression to compute Laplacian, thereby necessitating multiple back-propagation calls to compute each double derivative. Please refer to the supplementary code/models/losses.py line 37-53.
>
> **Q2** *How does the number of parameters of the methods in tab 1 compare to each other?*
>
> **->** See table below.
>
>
> | Model    | Parameters |
> |----------|------------|
> | IGR      | 2.1M       |
> | SAL      | 2.1M       |
> | IGR+FF   | 2.1M       |
> | PHASE+FF | 2.1M       |
> | SIREN    | 0.26M      |
> | DiGS     | 0.26M      |
> | Ours     | 0.20M      |

---

> > ### Comment · Reviewer_urXf · 2023-08-14
> >
> > The authors have sufficiently address my concers.

---

### Official Review · Reviewer_Nfwa · 2023-07-05

**Soundness:** 4 excellent
**Presentation:** 4 excellent
**Contribution:** 3 good
**Rating:** 8
**Confidence:** 3

**Summary:**

The paper first utilizes Von Neumann analysis to put forward a plausible explanation for instability of neural SDFs as the function space enlarges (as network parameters increase). Along the way, they show that some of the commonly used regularizers can combat this instability if they are weighted heavily enough. They develop a new regularizer based on the fact that the gradient is in the null space of the Hessian for an SDF (almost everywhere), and demonstrate its superior performance on three benchmark datasets. Lastly, they utilize quadratic layers instead of linear ones in their neural SDFs, a novel network structure for neural implicit representations, which seems to increase the expressivity of their SDFs.

**Strengths:**

- Brings in theory and analysis on gradient descent of functionals (on reduced basis function spaces), to shed light on the stability of the Eikonal loss and various regularizers already in use.
- Development of a novel regularizer that is more stable with respect to increasing parameter count.
- Demonstration of improvement in quality with use of quadratic layers, a novel architecture.
- The writing and presentation is quite clear and easy to follow.
- The insights from this article will help to evaluate the stability of future regularizers and inform their development.

**Weaknesses:**

- The paper could have provided a bit more context for the Von Neumann analysis. They provide a reference, but it still might be helpful to give a bit of an introduction to the analysis (or give a more specific chapter/section/theorem reference).
- The ablation study could have also been done on ShapeNet. In particular, I'd be curious to see the quadratic vs linear results.

**Questions:**

See the two points above under weaknesses.

**Limitations:**

Yes, the authors have adequately noted the limitations of their method. The main one is the increased training time for quadratic layers.

---

> ### Author Rebuttal · Authors · 2023-08-10
>
> **Weakness1** *The paper could have provided a bit more context for the Von Neumann analysis. They provide a reference, but it still might be helpful to give a bit of an introduction to the analysis (or give a more specific chapter/section/theorem reference).*
>
> **->** We will add more related references in the final version, e.g., see L. Evans/PDEs Chapter 4.3 on spectral methods, and Trefethen Chapter 4. We added a reference [33] that provides some tutorial of Von Neumann analysis in the context of DL; we will specify this more explicitly in the final version. Unfortunately, its a collection of techniques and not a single theorem.
>
> **Weakness2** *The ablation study could have also been done on ShapeNet. In particular, I'd be curious to see the quadratic vs linear results.*
>
> ->See Global Response.

---

> > ### Comment · Reviewer_Nfwa · 2023-08-15
> > **Evaluation Unchanged**
> >
> > Thank you for the responses. I am still quite positive on the paper, and am happy with the addition of further ablation experiments.

---

### Official Review · Reviewer_bsBS · 2023-07-06

**Soundness:** 3 good
**Presentation:** 3 good
**Contribution:** 3 good
**Rating:** 7
**Confidence:** 3

**Summary:**

The paper provides a theoretical framework to study the optimization of neural SDFs. In this framework, the eikonal loss - used as regularization to obtain a SDF - results unstable. Existing regularizers are proven to stabilize the eikonal loss, but they empirically lead to oversmoothing. To overcome this issue, the authors propose a novel regularization, promoting a small second derivative in the normal direction, to be used in combination with a more expressive network (i.e., quadratic layers). The experimental section provides comparison with sota in the SDF reconstruction task on three datasets.

**Strengths:**

In my opinion, the papers have several strengths.

*Importance of the task:* The optimization of neural SDFs is a significant task that finds applications in various pipelines. By addressing the stability and oversmoothing challenges, this paper contributes to improving the quality of SDF reconstructions, making it relevant to a wide range of domains.

*Practical Applicability:* The proposed regularizer can be easily integrated into existing SDF reconstruction pipelines, making it convenient for others to adopt in their work.

*Experimental Results:* The authors provide comprehensive experimental comparisons with state-of-the-art methods on three commonly used benchmark datasets. These results demonstrate the effectiveness of the proposed approach and support the assumptions made in the paper.

*Theoretical Justification:* The paper goes beyond empirical observations by providing a theoretical justification for the proposed regularization approach. This strengthens the credibility of the method and enhances the understanding of its underlying principles.

**Weaknesses:**

I think the main weaknesses to consider are the following.

*Increased Training Time:* The major weakness is the increased training time associated with the proposed method due to the use of quadratic layers, as noticed by the authors. The ablation study suggests that the regularizer alone does not improve performance, necessitating the use of a more expressive network. However, the more expressing network alone would lead to instability. This drawback should be taken into account when considering the practical implementation of the proposed approach.

*Limited Exploration of Neural Architectures:* It would have been valuable to explore alternative neural architectures that could potentially preserve high-frequency details without resorting to computationally expensive quadratic layers. Investigating other network settings, such as increasing the number of layers and neurons, could provide insights into potential trade-offs between computational efficiency and performance.

**Questions:**

It would be interesting to investigate how the proposed regularizer performs with different neural network configurations that do not require quadratic layers.

For instance, could increasing the number of layers and neurons provide a similar effect without significantly increasing computational costs?

Is the proposed framework utilizing sinusoidal activation functions?

Could positional encoding techniques be employed in combination with the proposed regularizer to enhance the representation capabilities of the network? This could potentially improve the network's ability to capture high-frequency detail and produce more accurate SDF reconstructions.

**Limitations:**

The authors adequately addressed the limitations.

---

> ### Author Rebuttal · Authors · 2023-08-10
>
> **Weakness1** *Increased Training Time: The major weakness is the increased training time associated with the proposed method due to the use of quadratic layers, as noticed by the authors. The ablation study suggests that the regularizer alone does not improve performance, necessitating the use of a more expressive network. However, the more expressing network alone would lead to instability. This drawback should be taken into account when considering the practical implementation of the proposed approach.*
>
> **->** As mentioned in Section 5.5, we believe a large part of the computational cost increase is due to the lack of a native Pytorch implementation of quadratic layers, which we currently implement with stacking linear layers.  We hope this work and other recent work with quadratic layers will motivate developers to produce a dedicated Pytorch implementation, which would address this current drawback.
>
> **Weakness2** *Limited Exploration of Neural Architectures: It would have been valuable to explore alternative neural architectures that could potentially preserve high-frequency details without resorting to computationally expensive quadratic layers. Investigating other network settings, such as increasing the number of layers and neurons, could provide insights into potential trade-offs between computational efficiency and performance*
>
> **->** The goal of this paper was to show the instability of the eikonal loss, how to mitigate it without over regularization, and show how this opens up the possibility to new, finer shape representations. Quadratic networks are one possibility that illustrates the latter contribution.  We hope that this paper simulates more thought in the community on networks structures to better represent shape.
>
> Having said that, a reason that we replace linear neurons with a quadratic neurons is that in FFN [17], it is shown that a standard MLP fails to learn high frequencies, despite increasing MLP parameters. One motivation for quadratic networks was to design a new network structure with more finer shape representation and the same or even fewer parameters. As the comment to Reviewer urXf Q2 indicates, we get a order of magnitude fewer parameters compared to FF(1/4 are due to quadratic and the rest are due to the use of sine activations).
>
> With regards to increasing number of layers and neurons, that has been explored extensively in the literature for quadratic [23] and extensively for linear MLPs, so the trade-off is well known. Again, we re-iterate our focus, which is to illustrate the possibility of finer shape representation network structures given that the key limitation, the instability of the eikonal loss, has been addressed in this work.
>
> **Q1** *It would be interesting to investigate how the proposed regularizer performs with different neural network configurations that do not require quadratic layers. For instance, could increasing the number of layers and neurons provide a similar effect without significantly increasing computational costs?*
>
> **->** See the previous response. We think the cost of quadratic networks will improve with customized GPU implementations in Pytorch. Increasing layers/neurons will increase the number of parameters, which is detrimental in size/power limited applications, e.g., edge applications.
>
> **Q2** *Is the proposed framework utilizing sinusoidal activation functions?*
>
> **->** Yes, we mentioned it in section 4.2.
>
> **Q3** *Could positional encoding techniques be employed in combination with the proposed regularizer to enhance the representation capabilities of the network? This could potentially improve the network's ability to capture high-frequency detail and produce more accurate SDF reconstructions.*
>
> **->** SIREN demonstrates that the sinusoidal activations are better than ReLU + P.E. when fitting the high-frequency signals. Typically one does not use sinusoidal activation and P.E. together. The combination of sinusoidal activation and P.E. is an interesting future topic, tangential to our focus in the present paper.

---

> > ### Comment · Reviewer_bsBS · 2023-08-14
> >
> > I thank the authors for their clarification and insights. After having read the rebuttal and other reviews, my positive evaluation of this work remains unchanged.

---

### Official Review · Reviewer_biQ8 · 2023-07-06

**Soundness:** 4 excellent
**Presentation:** 3 good
**Contribution:** 4 excellent
**Rating:** 7
**Confidence:** 5

**Summary:**

The paper gives a theoretical framework for analysis the stability of neural SDF optimisation, by considering the gradient descent optimisation as a PDE. They can then analysis the gradient descent on Eikonal loss PDE, and show that it is inherently unstable, and further show that normal loss or divergence loss (which are widely used in the literature to train these methods) help stabilise the PDE. Finally they explain why the divergence loss oversmoothes (as it is equivalent to mean curvature when the eikonal term is satisfied) so they introduce a new loss, Laplacian normal regularisation, that removes this issue by being faithful to SDFs but giving enough second order information to stabilise the PDE. To do this they take the derivative of both sides of the Eikonal equation, and enforce this constraint as a loss. They also introduce quadratic layers in the MLP, in order to fit better to curved surfaces. Their results show good improvements in surface reconstruction performance, however their ablation studies do not make it clear whether the improvement is mainly from quadratic layers or from the new regularisation term.

**Strengths:**

- Interesting theoretical framework for analysis the stability of neural SDF optimisation, and explanation for why normals or divergence loss helps with stability
- Great insight into why the divergence loss from DiGS is not optimal in terms of mean curvature
- Using second order information derived from the Eikonal equation makes sense

**Weaknesses:**

- The ablation results on SRB seem to show that quadratic layers could be more important than the Lapacian normal regularisation, but it is hard to tell as the overall improvements are minor. It would be better to run the ablation on ShapeNet, where there is a much larger improvement in the full model compared to DiGS.
- If the motivation of quadratic layers is for a better fit to curved surfaces, then a simpler alternative with a lot less parameter increase would be to change the input to quadratic basis functions, e.g. [x,y] -> [1,x,y,xy,x^2,y^2]. Have the authors tried this, or is there more motivation for quadratic layers?

Overall the theoretical analysis is a great contribution. While the results are strong (small improvement on SRB but SRB is close to saturated already so large improvements are hard, and there are impressive improvements on ShapeNet), the ablation study being on SRB doesn't make it clear how much of the contribution is from the main point of the paper, Lapacian normal regularisation, or from quadratic layers. Doing the ablation study on ShapeNet would clear this up.

**Questions:**

- How important is the annealing strategy on Laplacian normal regularisation? In DiGS, the motivation for the annealing strategy was that the divergence loss, while very "stabilising", caused oversmoothing, hence the need to anneal the loss. Your analysis explains that the oversmoothing is due to the mean curvature being penalised everywhere which is not ideal, and instead introduce Lapacian normal regularisation to mitigate this, so it sounds as if annealing would not be necessary?
- In equation 15/16, it should be a matrix-vector product not a dot product, i.e. D^2u(x)\nabla u(x) rather than D^2u(x)\cdot \nabla u(x).

**Limitations:**

- Some discussion of limitations w.r.t. DiGS (I guess they assume the reader understands they inherit the limitations of DiGS, e.g. the extra time required for computing and backpropagating through second derivatives, which makes sense)
- Would like some discussion of whether the author believes that this new loss fixes the issues with thin features

---

> ### Author Rebuttal · Authors · 2023-08-10
>
> **Weakness1** *The ablation results on SRB seem to show that quadratic layers could be more important than the Laplacian normal regularisation, but it is hard to tell as the overall improvements are minor. It would be better to run the ablation on ShapeNet, where there is a much larger improvement in the full model compared to DiGS.*
>
> **->** See Global Response.
>
> **Weakness2** *If the motivation of quadratic layers is for a better fit to curved surfaces, then a simpler alternative with a lot less parameter increase would be to change the input to quadratic basis functions, e.g. $[x,y] -> [1,x,y,xy,x^2,y^2]$. Have the authors tried this, or is there more motivation for quadratic layers?*
>
> **->** The composition of multiple quadratic layers can approximate any desired order of a Taylor series and the composition of multiple linear layers is still only a linear function. Thus, if the input to a linear network is the basis that the reviewer mentions, the output is still quadratic not a higher order polynomial as we desire. This is discussed in the first paragraph of section 4.2.
>
> **Q1** *How important is the annealing strategy on Laplacian normal regularisation? In DiGS, the motivation for the annealing strategy was that the divergence loss, while very "stabilising", caused oversmoothing, hence the need to anneal the loss. Your analysis explains that the oversmoothing is due to the mean curvature being penalised everywhere which is not ideal, and instead introduce Laplacian normal regularisation to mitigate this, so it sounds as if annealing would not be necessary?*
>
> **->** The second order loss corresponds to 4th order PDE in the gradient optimization, which involves a diffusion term that requires small learning rates for stability. The relatively large regularization at the start allows the optimization to enforce the constraint, which must be greater than the Lagrange multiplier, and avoid local optimizers.  The smaller regularization towards the end of the evolution allows for larger learning rates that speeds the convergence without being trapped by local optimizers if one employed a uniformly small regularization. Thus, while the annealing strategy may not be necessary to obtain accurate results, it does speed the convergence of the method.  Moreover, the directional Laplacian minimization does still retain some minor remnant of shape regularization only due to the finite parameterization network representation of the shape where a small amount of diffusion can occur away from the normal direction.  Thus, the annealing helps with recovering finer shape details (although much less so than needed for the original Laplacian regularization).
>
> **Q2** *In equation 15/16, it should be a matrix-vector product not a dot product, i.e. $D^2u(x)\nabla u(x)$ rather than $D^2u(x)\cdot \nabla u(x)$.*
>
> **->** We have used operator theoretic notation where the dot notation means "the operator $D^2u(x)$ applied to $\nabla u$", which corresponds to matrix vector multiplication in this case. We will add a comment.
>
> **Limitation1** *Some discussion of limitations w.r.t. DiGS (I guess they assume the reader understands they inherit the limitations of DiGS, e.g. the extra time required for computing and backpropagating through second derivatives, which makes sense)*
>
> **->** No comment necessary.
>
> **Limitation2** *Would like some discussion of whether the author believes that this new loss fixes the issues with thin features*
>
> **->** Penalizing the mean curvature as in the divergence loss can result in pinching thin structures and necks in the gradient flow, which is a well-known property of mean curvature. Since we are not penalizing the mean curvature of the surface, our approach has less of a problem with thin structures as we have seen in visual results.  As discussed in Reviewer biQ8 Q1, the flow can still have a small amount of diffusion in other (tangent) directions, so it may not fully "fix" the issue, though works well in practice.  We will add this into discussion.

---

> > ### Comment · Reviewer_biQ8 · 2023-08-18
> > **My concerns have been addressed, increasing my score from 6 to 7.**
> >
> > Thanks for the ablation results on ShapeNet, I think it is a very useful result for the paper, and properly distinguishes the main contribution from the quadratic layers. The results suggest that the L.n. loss improves the IoU (consistently getting whats inside and outside) while quadratic layers improves the Chamfer distance. This makes sense, quadratic layers are about better fit to the surface, while the L.n. loss is about regularising towards the correct shape.I hope the authors include this in the main part of their main paper.
> >
> > The authors have also sufficiently addressed my other concerns. Since I was already very positive about this paper with only the now given ablations results being the issue (and the results on that being in line with what I hoped/expected), I am raising my rating from a 6 to a 7.

---

### Official Review · Reviewer_Bytg · 2023-07-07

**Soundness:** 4 excellent
**Presentation:** 3 good
**Contribution:** 4 excellent
**Rating:** 6
**Confidence:** 4

**Summary:**

This paper presents a novel approach to learning implicit neural representations (INR) of shapes. The paper tries to address the issue of instability in optimizing neural signed distance functions (SDFs) using the eikonal loss, which can lead to irregularities in the reconstructed surfaces and convergence to sub-optimal solutions. The paper provides a theoretical framework based on geometric partial differential equations (PDEs) to analyze the eikonal loss and its instabilities.  They introduce a new Laplacian normal regularization that stabilizes the optimization without over-smoothing the surfaces.
Furthermore, the paper proposes the use of quadratic layers in neural SDFs, which allow for higher-order polynomial approximations of shape and capture finer details. The paper validates the approach through extensive experiments on benchmark datasets, showcasing the superiority of their method in reconstructing shapes compared to state-of-the-art methods.

**Strengths:**

- The paper demonstrates good writing skills, presenting complex concepts and theories in a clear and understandable manner.

- The paper provides strong theoretical explanations for the optimization of neural signed distance functions. By using geometric PDEs, the authors analyze the eikonal loss and its instabilities, revealing the underlying mechanisms and challenges in optimizing neural SDFs.

- Building upon the theoretical insights, the authors propose a new regularization term that addresses the instability issue without over-smoothing the surfaces.

**Weaknesses:**

- The paper states that their method improves the representation of shapes compared to state-of-the-art methods. However, the evaluation on the surface reconstruction benchmark dataset does not demonstrate significant performance improvements over other methods.

- The paper introduces a new regularization term to address the instability issue in optimizing neural signed distance functions (SDFs). However, it does not include ablation experiments to investigate the impact of different hyperparameters of the loss function. A comprehensive analysis of the regularization term's sensitivity to hyperparameters could provide valuable insights into its effectiveness and guide the selection of optimal settings.

**Questions:**

- What are the differences between the linear backbone and Digs? Why does $L_{div}$ perform slower than $L_{L.n.}$ in terms of time efficiency?

- Why does the proposed method show limited improvement on the Surface Reconstruction Benchmark dataset? Can you provide some explanations or insights into this observation?

**Limitations:**

Limitations of the proposed method have been discussed in a sub-section of the paper.

---

> ### Author Rebuttal · Authors · 2023-08-10
>
> **Weakness 1** *The paper states that their method improves the representation of shapes compared to state-of-the-art methods. However, the evaluation on the surface reconstruction benchmark dataset does not demonstrate significant performance improvements over other methods.*
>
> **->** Since SRB is less challenging than the other datasets, without many thin and/or complex structures, DiGS already performs well. Nevertheless, noting that the metrics for SRB represent the average of 5 shapes in the dataset, we go on to offer a finer inspection in the supplementary by providing the performance for each individual shape. DC, Garagoyle and LordQuas  are close to saturated already so large improvements are hard. However, there is a significant improvement for Anchor and Daratech. Especially for the visual result of the Anchor, you can clearly see our method fixes the hole reconstruction.
>
> **Weakness 2** *The paper introduces a new regularization term to address the instability issue in optimizing neural signed distance functions (SDFs). However, it does not include ablation experiments to investigate the impact of different hyperparameters of the loss function. A comprehensive analysis of the regularization term's sensitivity to hyperparameters could provide valuable insights into its effectiveness and guide the selection of optimal settings.*
>
> **->** We have conducted this experiment on SRB varying regularization weights. It shows that around the optimal weight choice, the results are not sensitive.  Furthermore, increasing the weight beyond the optimal, only degrades results slightly since that simply enforces further a constraint that is true of all SDFs, without smoothing the geometry much.  This is consistent Lagrange multiplier theory for enforcing a constraint into the optimization problem.
>
> |            | GT    |       | Scans         |               |
> |-----------|-------|-------|---------------|---------------|
> | $\alpha_l$ | $d_C$ | $d_H$ | $d_{\vec{C}}$ | $d_{\vec{H}}$ |
> | 10         | 0.264 | 6.089 | 0.099         | 1.513         |
> | 50         | 0.191 | 3.799 | 0.096         | 1.485         |
> | $100^*$    | **0.180** | **2.800** | **0.096**         | **1.453**         |
> | 200        | 0.188 | 3.520 | 0.097         | 1.495         |
> | 300        | 0.192 | 3.497 | 0.102         | 1.535         |
> | 400        | 0.187 | 3.177 | 0.102         | 1.537         |
> | 500        | 0.194 | 3.557 | 0.098         | 1.499         |
> Table: Varying $\alpha_l$ and performance. The relationship between $\alpha_l$ and the performance is not salient. We may mention that the weight needs to be tuned based on different tasks, but a relatively larger weight is preferred given the annealing strategy.
>
> **Q1.1** *What are the differences between the linear backbone and Digs?*
>
> **->** The DiGS result presented in Table 1 is the result reported in their original paper. To show the improvement of each component, we run all experiments using the same experiment setting(hyperparameter, environment, GPU). $Lin + L_{div}$ is the result of DiGS we got.
>
> **Q1.2** *Why does $L_{div}$  perform slower than $L_{l.n.}$  in terms of time efficiency?*
>
> **->** The major savings comes from the number of back-propagation calls.  In our approach, we only need to compute derivatives of scalar functions in comparison to the divergence term. Since $D^2(x)\nabla u(x)$ can be computed conveniently as half the derivative of $\langle \nabla  u(x), \nabla  u(x) \rangle$ with respect to $x$, we can compute $L_{L.n} = \langle D^2(x)\nabla u(x), \nabla u(x)\rangle$ by a single back-propagation call. In contrast, there is no simpler expression to compute Laplacian, thereby necessitating multiple back-propagation calls to compute each double derivative. Please refer to the supplementary code/models/losses.py line 37-53.
>
>
> **Q2** *Why does the proposed method show limited improvement on the Surface Reconstruction Benchmark dataset? Can you provide some explanations or insights into this observation?*
>
> **->** See Weakness1.

---

> > ### Comment · Reviewer_Bytg · 2023-08-20
> > **Official Comment by Reviewer Bytg**
> >
> > After reading the rebuttal and other reviews, the response has resolved my concerns.

---

### Author Rebuttal · Authors · 2023-08-10

We thank the reviewers for a thorough review. There are some concerns mentioned by several reviewers so we will address them here.

**[Slight Correction on ShapeNet Results]**: We apologize that there are some minor errors in the evaluation of ShapeNet in the initial submission. We mistakenly put the result of one category on the table, not the overall result. We have fixed them and updated the new result for ShapeNet here. It does not affect the theory and conclusion of our paper.

**[Ablation Study on ShapeNet]**: In addition, some reviewers requested an ablation study of ShapeNet, so we include it here.  Results are reported on the table below. They show that on 5 out of 6 metrics, our normal Laplacian regularization using linear networks outperforms DiGs (optimized hyper-parameter), showing the utility of the new regularization. On 4 out of 6 metrics, our normal Laplacian regularization out-performs the standard Laplacian regularization using quadratic networks, again showing the utility of our new regularization separately. Note that in both cases the metrics where the normal Laplacian normal performs worse are just slightly worse compared to the amount of increase in the other metrics.
On all 6 metrics, the quadratic network using the same Laplacian regularization as DiGs out-performs DiGs, showing the utility of quadratic networks alone. Note that each of our contributions, i.e., Laplacian normal regularization and quadratic layers, each separately show increase performance against DiGs (except one metric in the linear case) even though the hyper-parameters were not optimized in the approaches compared against DiGs.


|                       |             | IoU $\uparrow$ |             |              | squared |Chamfer $\downarrow$              |
|-----------------------|-------------|----------------|-------------|--------------|------------------------------|--------------|
| Method                | mean        | median         | std         | mean         | median                       | std          |
| DiGS                  | 0.9390      | 0.9764         | 0.1262      | 1.32e-4      | 2.55e-5                      | 4.73e-4      |
| Lin+$L_{\text{L. n.}}$ | 0.9586      | 0.9809         | 0.0993      | 1.71e-4      | 1.23e-5                      | 1.20e-3      |
| Qua+$L_{\text{div}}$  | 0.9593      | 0.9852    | 0.1130      | 5.45e-5 | 1.05e-5                      | 3.60e-4      |
| Ours                  | 0.9671 | 0.9841         | 0.0878 | 6.86e-5      | 6.33e-6                 | 3.34e-4 |
Table: Ablation study on ShapeNet.

We also named our method StEik(Stable Eikonal) for future convenient reference.

---

### Decision · Program_Chairs · 2023-09-21

**Decision:**

Accept (poster)

**Comment:**

This paper presents an analysis of the popular Eikonal loss in terms of evolution PDE stability. After identifying a potential instability in the standard formulation this paper suggests a novel regularization improving upon this aspect. The paper presents experiments testing this new stabilization technique and other architecture changes (e.g., quadratic layers), comparing also to relevant previous works. The paper is well written, presents a novel and useful analysis and is able to justify its theoretical observations empirically. The paper demonstrates a rather mild improvement qualitatively and quantitatively over baselines. Please incorporate the new ablations in the camera ready version of the paper and improve exposition as requested by reviewers.